# The Sustainable Future of Carbon Farming with Virginia Fanpetals, a Carbon-Negative Species

**Roman Molas** [1,*], **Halina Borkowska** [2,†], **Dominika Skiba** [2], **Barbara Sawicka** [2] and **Ewa Skulimowska** [1]

1    Research Unit, USida R&D, ul. Czardasza 12, 02-169 Warszawa, Poland; sqooliewa@gmail.com
2    Department of Plant Production, Technology and Commodity Sciences, University of Life Sciences in Lublin, ul. Akademicka 13, 20-033 Lublin, Poland; halina.borkowska@up.lublin.pl (H.B.); dominika.skiba@up.lublin.pl (D.S.); barbara.sawicka@up.lublin.pl (B.S.)
*    Correspondence: usida@wp.pl
†    Deceased.

**Abstract:** In 2016–2018, an experiment with *Sida hermaphrodita* L. Rusby (Virginia fanpetals) called Sida was conducted in the system of random blocks on sandy and sandy loam soil. Each year, the yields of dry aboveground mass and dried mass of roots were determined. Before sowing Sida and after the end of the research, the composition of earthy fractions in the soils was determined. In the stalks and roots of Sida, the percentage content was determined, and the mass of carbon was calculated. When Sida was grown in sandy soil, the average dry mass yield (stalks and roots) in the third year of cultivation (2018), was slightly above 13 $t \cdot ha^{-1}$, and in the sandy loam soil it was three times higher (36 $t \cdot ha^{-1}$). In the third year of cultivation, the carbon sequestration in the stems and roots exceeded 5 $t \cdot ha^{-1}$ (sandy soil), and 15 $t \cdot ha^{-1}$ (sandy loam). The carbon content in the stems was 46.25%, and in the roots–42.67%. The results were statistically processed using ANOVA. It was found that the cultivation of Sida caused a percent decrease in the sand fraction, an increase in the floatable (fine) fraction in sandy soil, and an increase in humus content; the opposite changes were noted in the sandy-loam soil.

**Keywords:** Virginia fanpetals; carbon farming; $CO_2$ sequestration; humus; carbon-negative species





## 1. Introduction

Carbon dioxide emissions into the atmosphere continue to increase. The global forest area, forest being the main "consumers" (sequestrants) of this gas, is systematically decreasing, according to various data, by about five million hectares annually [1]. Although the forest area in Poland is systematically growing, their ability to sequester $CO_2$ is slow and depends on age. An additional negative effect is the disappearance of coniferous forests, which assimilate throughout the year (in Poland and Europe), caused by a huge invasion of the European spruce bark beetle (*Ips typographus*). Consequently, the growing interest in field plant species capable of rapid and intensive use of carbon dioxide for biomass is growing. Carbon sequestration in the soil in the form of root biomass is important.

Research on Sida in Poland has been done for the last 40 years; most of it was performed at the University of Life Sciences in Lublin [2–6]. In addition to the yield of aboveground mass, this species-Sida-develops an extensive, strong root system. A lot of attempts are also being conducted in other European countries and the USA [7,8]. Sida is native to USA, and is listed as a rare and endangered species.

The assumed research hypothesis determines that Sida, in field cultivation, is capable of carbon sequestration in the root system and in the aboveground mass and has a positive effect on the soil granulometric and physicochemical composition; unlike the alternative hypothesis that the cultivation of this species does not significantly affect the above-mentioned features. Hence, a three-year study of the Sida cultivation was conducted to determine which of the hypotheses was true.

## 2. Materials and Methods

Over the course of 2016–2018, an experiment was conducted with Sida, on sandy, and sandy loam soil, at an individual farm in Brzeziny in the Świętokrzyskie Voivodeship (21°00′ E; 50°36′ N, 300 m above sea level). The experiment was performed using the randomized block method in triplicate. The factor of the experiment was the type of soil: (a) sandy soil, or (b) sandy loam soil. The basic nutrients detected at the government laboratory in Kielce, in both soils, were as follows: 1.4 P; 2.2 Mg; 4.2 K mg in 100 g of soil samples. In both soils, pH was determined as 5.1. The area of plots to be harvested was 25 m$^2$. The plots were selected randomly from two neighboring plantations of Sida, with a total area of four hectares.

### 2.1. Agri-Technical and Rainfall Conditions

On May 10, 2016, seeds were sown on four hectares, at a quantity ensuring a density of 70,000 plants per hectare. In the first year of cultivation, no fertilizers were used. Weeds were removed mechanically. No plant protection (pesticides) was applied. The harvest was collected by hand in plots, and the weight of the samples (all stems from each randomly selected plot of twenty-five m$^2$ were collected) was determined immediately after the harvest in the field. In the second and third years, based on soil analysis, mineral fertilization was applied, amounting to 100 kg of nitrogen, 50 kg of phosphorus and 90 kg of potassium, as pure elements, per hectare. The total rainfall in 2016 was 234 mm plus 533 mm of winter snow and rain; in 2017 it was 400 mm plus 614 mm of winter snow and rain; and in 2018 it was 355 mm plus 512 mm of winter snow and rain.

### 2.2. Soil Sampling

Soil samples were collected simultaneously with the excavation of the roots, each year after the end of plant vegetation, at the end of November. Soil samples were taken separately from two different plantations, collected from a layer 1 cm to 30 cm below the surface, then mixed, and sent as one kg samples to the recommended laboratory. The aboveground plant material from each plot was the same (all the shoots from the plot), was weighed and then chopped, mixed and a two kg sample was then weighed and sent to the certificated laboratory for analysis.

The chemical and physicochemical properties of the soil were determined at the certified laboratory of the Regional Chemical and Agricultural Station in Kielce (Scope of Accreditation for Testing Laboratory Nr/No AB 333), the determination of the soil granulometric composition was done according to laser diffraction [9,10]; and organic carbon content—Corg.—was determined by the Tiurin method [11,12]. Due to its simplicity, the Tiurin method is commonly used in the assessment of organic matter content in cohesive soils [11].

### 2.3. Sampling of Roots for Testing and Determination of Dry Matter

Roots from each twenty-five m$^2$ plot were excavated manually from a layer up to 30 cm deep, then weighed and cut into 10 cm pieces. All the pieces were then mixed, and two kg of the root mixture from each plot was assigned for analysis. The rest of the cut pieces of the roots were planted in rows to restore the planting density. The roots were harvested at the end of November. Dry matter was determined in the roots using the drying method, following which the samples were ground and intended for chemical analyses [13,14].

### 2.4. Sampling of Stems and Determination of Dry Matter

After harvesting the aboveground mass, 10% of the stems were sampled from each plot. The dry matter content in the stems was determined using the drying method in the laboratory at the Regional Chemical and Agricultural Station in Kielce [14], which is located closest to the experimental field. On this basis, the dry weight of stems and roots was calculated per unit area, converted to tons per hectare.

## 2.5. Carbon Determination in Plant Material

At the Institute of Power Engineering in Warsaw, the carbon content in the Sida stems was determined using the FTIR Gasmet DX-4000 analyzer (Gasmet Technologies Oy, Vantaa, Finland). The carbon content in the roots was determined at the Wood Technology Institute in Poznań using the FlashEA 1112 NC Analyzers (Thermo Fisher Scientific, Waltham, MA, USA). Based on these determinations, the amount of carbon bound by Sida was calculated.

## 2.6. Statistical Calculations

The statistical processing of the results was performed using the variance model (ANOVA) (SAS 9.2 2008) [15]. The significance of the sources of variation was verified with the Fischer–Snedecor "F" test, while the value of LSD0.05 was estimated using the Tukey test [16]. T-Tukey's multiple comparison tests enabled a detailed comparative analysis of means by isolating and designating statistically homogeneous groups of means (homogeneous groups). The calculated *p* values determine the significance and extent of the influence of the studied factors on the differentiation of the results of the analyzed variables, comparing them with the most commonly accepted alpha significance levels (0.05). The averaged letter indices define the so-called homogeneous (statistically homogeneous) groups. The presence of the same letter index with means (at least one) indicates that there is no statistically significant difference between them [17].

## 3. Results

### 3.1. Biomass Yields

The results presented in Table 1 show a significant increase in Sida biomass in the preliminary three years of cultivation on sandy and sandy loam soil.

**Table 1.** Sida aboveground yield in oven dry tons (ODT).

| Year | Type of Soil | | |
|:---:|:---:|:---:|:---:|
| | **Sandy** | **Sandy Loam** | **Mean** |
| 2016 | 1.115a | 0.218a | 0.667a |
| 2017 | 1.602a | 8.780b | 5.191b |
| 2018 | 3.190b | 15.760c | 9.475c |
| LSDp$_{0.05}$ for years × type of soils */ for years ** | 0.765 * | | 0.383 ** |
| Mean | 1.968a | 8.253b | 5.111 |
| LSDp$_{0.05}$ for type of soils *** | 0.255 *** | | |

The same letters (a, b, c) in the table mean there is no statistically significant difference. *, **, ***—refer to the LSD values in the following columns: *—LSDp0.05 for years × type of soils; **—LSDp0.05 for years; ***—LSDp0.05 for type of soils.

From 1.1 t·ha$^{-1}$ in the first year to almost three times higher yields in the third year. The yields of the aboveground mass are low, as they are determined by the perennial species development rate, and their values over subsequent years are statistically significantly different. It should be noted here that the aboveground mass of Sida, which is perennial, completes its growth and development due to the end of the growth season, and is harvested each year. The formation of the root mass is completely different (Table 2). It accumulates reserve substances, which determine the long-term durability of the plantation and annual higher yields of aboveground mass.

**Table 2.** Roots yield in ODT per hectare.

| Year | Type of Soil | | |
|---|---|---|---|
| | **Sandy** | **Sandy Loam** | **Mean** |
| 2016 | 3.363a | 3.160a | 3.262a |
| 2017 | 7.007b | 10.367b | 8.687b |
| 2018 | 9.956c | 19.890c | 14.932c |
| LSDp$_{0.05}$ for years × type of soils */ for years ** | | 1.737 * | 0.869 ** |
| Mean | 6.775a | 11.139b | 8.957 |
| LSDp$_{0.05}$ for type of soils *** | | 0.579 | |

The same letters (a, b, c) in the table mean there is no statistically significant difference. *, **, ***—refer to the LSD values in the following columns: *—LSDp0.05 for years × type of soils; **—LSDp0.05 for years; ***—LSDp0.05 for type of soils.

In the top 30 cm of the soil, Sida developed over three tons of roots per hectare in the first year alone. In the third year of cultivation, the root weight increased threefold. It is worth noting that the three-year root mass is also three times greater than the yield of the aboveground part in the third year. The increase in root mass (as the roots are not dug up every year) reflects the tendency characteristic for perennials. As in the case of stems, there are statistically significant differences between the values of the root weight in individual years.

### 3.2. Carbon Sequestration

A similar tendency to the increase in plant mass is visible in the increase in the mass of stored carbon (Table 3). The experience factor significantly differentiated the value of this feature, both for the mass of shoots and the mass of roots. The results from specific years showed statistically significant differences.

**Table 3.** Total stem and root yield in ODT per hectare.

| Years | Type of Soil | | |
|---|---|---|---|
| | **Sandy** | **Sandy Loam** | **Mean** |
| 2016 | 4.478a | 3.378a | 3.928a |
| 2017 | 8.609b | 19.147b | 13.878b |
| 2018 | 13.146c | 35.650c | 24.398c |
| LSDp$_{0.05}$ for years × type of soils */ for years ** | | 2.196 * | 1.098 ** |
| Mean | 8.744a | 19.392b | 14.068 |
| LSDp$_{0.05}$ for type of soil *** | | 0.732 *** | |

The same letters (a, b, c) in the table mean there is no statistically significant difference. *, **, ***—refer to the LSD values in the following columns: *—LSDp0.05 for years × type of soils; **—LSDp0.05 for years; ***—LSDp0.05 for type of soils.

In the third year of cultivation, almost 1.5 t·ha$^{-1}$ carbon was bound in the stems, and the sequestration of $CO_2$ in the roots amounted to over four tons of carbon per hectare of this element (Table 4).

**Table 4.** Carbon sequestration in stalks of Sida in C/ha (tons).

| Year | Type of Soil | | |
|---|---|---|---|
| | **Sand** | **Sandy Loam** | **Mean** |
| 2016 | 0.516a | 0.124a | 0.320a |
| 2017 | 0.741a | 4.061b | 2.401b |
| 2018 | 1.475b | 7.280c | 4.382c |
| LSDp$_{0.05}$ for years × type of soils */ for years ** | | 0.369 * | 0.184 ** |
| Mean | 0.911a | 3.825b | 2.368 |
| LSDp$_{0.05}$ for type of soil *** | | 0.123 *** | |

The same letters (a, b, c) in the table mean there is no statistically significant difference. *, **, ***—refer to the LSD values in the following columns: *—LSDp0.05 for years × type of soils; **—LSDp0.05 for years; ***—LSDp0.05 for type of soils.

In the sandy loam soil, the yield of Sida grown from seeds is also low in the first year of cultivation (a feature of perennials).

The dry weight yields of stems harvested in the first year on sandy loam soil presented in the table amounted to slightly over 0.2 t ha$^{-1}$, yet in the second year they were many times higher at 8.7 t ha$^{-1}$. The amount of dry matter of roots in the first year of cultivation in sandy loam soil is similar to that in sandy soil. In the second year, the root weight of Sida in sandy loam is slightly higher than that of sandy soil in the third year of cultivation and show statistically significant differences.

Significant carbon sequestration is associated with a large root mass.

More than four tons of carbon per hectare (Table 5) were already achieved in the second year of cultivation, in the root system located in the top 30 cm layer of sandy loam soil, from Sida bound in the root system.

**Table 5.** Carbon sequestration in roots of Sida in C/ha (tons).

| Year | Type of Soil | | |
|---|---|---|---|
| | **Sandy** | **Sandy Loam** | **Mean** |
| 2016 | 1.435a | 1.348a | 1.392a |
| 2017 | 2.990b | 2.990b | 3.707b |
| 2018 | 4.249c | 4.249c | 6.368c |
| LSDp$_{0.05}$ for years × type of soils */ for years ** | | 0.567 * | 0.283 ** |
| Mean | 2.891a | 4.753b | 3.822 |
| LSDp$_{0.05}$ for type of soils *** | | 0.198 *** | |

The same letters (a, b, c) in the table mean. there is no statistically significant difference. *, **, ***—refer to the LSD values in the following columns: *—LSDp0.05 for years × type of soils; **—LSDp0.05 for years; ***—LSDp0.05 for type of soils.

However, in sandy soil, this result was achieved one year later, which shows the effect of soil abundance in water (fertilization was the same).

It is worth paying attention to the large mass of roots obtained from only 30 cm of the top layer. Despite this, Sida produces a larger root mass, as the root system reaches more than 350 cm into the soil, and the main mass is in the top 25–50 cm layer.

Table 6 shows the total results of carbon sequestration in Sida shoots and roots over the three years of research cultivation.

**Table 6.** Total results of carbon sequestration in Sida shoots and roots in C/ha (tons).

| Year | Type of Soil | | |
|---|---|---|---|
| | Sandy | Sandy Loam | Mean |
| 2016 | 1.951a | 1.472a | 1.712a |
| 2017 | 3.732b | 8.484b | 6.108b |
| 2018 | 5.724c | 15.776c | 10.750c |
| LSDp$_{0.05}$ for years × type of soils */ for years ** | | 0.927 * | 0.465 ** |
| Mean | 3.802c | 8.577b | 6.190 |
| LSDp$_{0.05}$ for type of soils *** | | 0.309 *** | |

The same letters (a, b, c) in the table mean there is no statistically significant difference. *, **, ***—refer to the LSD values in the following columns: *—LSDp0.05 for years × type of soils; **—LSDp0.05 for years; ***—LSDp0.05 for type of soils.

The two most pronounced relationships, which are statistically significant and have the same fertilization and rainfall, result from the difference in the greater capacity of sandy loam to store water than sandy soil. and the huge and rapid development of the plant and roots, which progresses with age.

Undoubtedly. the large mass of perennial roots. and several tons of fallen leaves each year, significantly change the soil-biohabitat. An example of this could be the results presented in Table 7.

**Table 7.** Soil environment effect of the cultivation of Sida on the content of earthy fractions (<2 mm) in sandy and sandy loam soil.

| Agronomic Category of Soil | the Term of the Marking | Percentage of Fractions | | |
|---|---|---|---|---|
| | | Sand (0.05–2.00) | Silt (0.05–0.002) | Loam (<0.002) |
| | | mm | | |
| Sandy soil | Before cultivating | 76.55 | 21.45 | 2.00 |
| | After three years of cultivation | 72.33 | 22.95 | 4.72 |
| Sandy loam | Before cultivating | 53.31 | 43.30 | 3.39 |
| | After three years of cultivation | 56.73 | 38.67 | 4.60 |

After three years of cultivation, the improvement in soil quality in both cases is related to the 1% increase in the content of humus in the sandy soil (the source is the falling Sida leaves) and the improvement of the water–air ratio in sandy loam soil.

The results of the soil particle size analysis are presented in Table 7. The experiment was conducted in two types of soil: sandy loam soil and medium loam soil. In terms of the percentage of sand, silt and clay fractions, the first type of soil is the granulometric subgroup-light loamy sand. The share of sand here was 76.55%, silt 21.45%, and loam 2.00%. In terms of agricultural suitability, these soils belong to the slightly acidic rye complex, with an agronomic category of average soil [11,12]. After 3 years, where an experiment was conducted with a perennial plant with a high biomass yield, the content of the sandy fraction decreased, and the content of silt and loam slightly increased. However, this did not change the agronomic category of the soil. The root system of Sida in sandy soil, where sand sticks to the humus, because the level of humus increases due to the mass of falling leaves, causes the roots to crush the soil lumps in which the sand is stuck, hence changing its content.

In the experiment located in the second type of soil, according to the percentage of sand, silt and loam fractions, where the sand fraction was 53.31%, the silt fraction was

43.3% and the loam fraction was 3.39%, which corresponds to the composition of medium loam, this soil is classified as heavy soil in the agronomic category. After 3 years, some changes in the soil particle size composition were observed, especially a decrease in the silt fraction, but these were not significant changes and did not lead to a change in the agronomic category of the soil.

The content of fine sand fraction increased significantly (by 2.72 points). A similar increase (statistically insignificant) was recorded in the silt content. This is especially important in light soil with a low water-holding capacity. Due to the increase in the content of humus in the soil (a dozen tons of fallen leaves annually, per hectare), sandy soil susceptible to drought, obtained a significant water retention capacity, similar to sandy loam soil.

## 4. Discussion

The vast majority of studies on the carbon sequestration of perennial energy crops concern grasses and trees (*Miscanthus* sp., *Salix* sp., *Populus* sp.). It is therefore difficult to compare the results of our work on dicotyledonous perennial with monocotyledon [18] perennials (another type of photosynthesis), measured with different methods and even at different depths, or even less with trees [19], or annual plants [20].

The research conducted so far in Poland (and in Europe) concerned mostly the yield of the aboveground Sida biomass. There is not a single study in which the carbon sequestration in Sida biomass would be investigated, considering the yield of underground biomass and the impact of several tons of fallen leaves per annum, feeding the humus and soil organic carbon (SOC) content in the soil.

According to the guidelines of the European Commission a drop in the humus content below 3.5% may indicate desertification of a given area and the need to introduce appropriate corrective measures in the cultivation of arable land and plants. If we take this criterion, the problem of desertification concerns most soils in Poland, as the average content of humus in soils is assumed to be about 1.9%. The division of soils in Poland according to the humus content is assumed to be poor soils containing 0.1% to 1% of humus; low, from 1% to 2%, medium, from 2% to 4%; and fertile, above 4% of humus.

The results of our research clearly show the advantages of Sida cultivation in terms of increasing the humus and carbon content while improving soil quality (Table 7).

Previous research by the University of Life Sciences in Lublin showed the beneficial effect of Sida cultivation on the soil. The studies carried out in sewage sludge (sludge layer 50–60 cm thick, spread over the soil) showed that after several years of cultivation of Sida, a clumpy structure, called "peds" (soil micro aggregates) formed in the originally muddy mass of the sludge [21,22]. Other species, including *Salix* and *Miscanthus*, have dried 100%. Several years of cultivation of the Sida species as a fore crop allowed the cultivation of other plants in this changed sediment [23], including wheat, with good yields from a quantitative and qualitative perspective. The structure-forming effect of Sida on the sediment substrate, which was the sewage sludge, has long raised the question, whether the cultivation of Sida on sandy soils is also able to improve their quality. Hence, the idea of carrying out the research which is the subject of the current manuscript.

Sida, like many other perennial species, reaches the full yield in the 3rd–4th or even 5th year of cultivation; it depends mainly on soil fertility and weather conditions. In light soil with paltry nutrient content and low water retention capacity, the yield of stems is irrelevant in the initial cultivation period. Along with the significantly growing root system, in the second year, they oscillate between two t·ha$^{-1}$ and more; however, in the fifth year, they exceed 11 t per hectare [24]. In the results presented above, in the initial period of cultivation, the yields of energy biomass were also low on sandy soil. From 1.6 t in the second year to 3.2 t·ha$^{-1}$ in the third year. The full yield can be expected in the 4th–5th year of cultivation. Much higher yields were achieved on sandy loam soil. Despite a very small weight of stems in the first year, almost 8.8 t ha$^{-1}$ were harvested in the second cultivation

year. In the literature on the subject, you can find related results (10.4 t per hectare) for the second year of Sida cultivation [25,26].

The content of carbon in the dry root matter is over 4%, slightly lower than in the aboveground energetic mass. The high-carbon content in a large mass of roots contributes to a significant amount of this element being stored and sequestered for years (plantation durability/lifespan is estimated at 20–30 years) under the soil surface.

Very rarely, attention is paid to plant residues, which also contain carbon but are usually removed from the field. There are no crop residues in the Sida cultivation as 100% of the aboveground biomass and carbon are harvested [5].

Our results [5,27], indicating a large share of roots, and a huge role of the roots in carbon sequestration, were noticed earlier, although in other species [28]. Some research teams have already noticed the environmental benefits of growing energy perennial [29].

The carbon contained in the stems from the first year in sandy loam, and on sandy soil also from the second year of cultivation, can be added to this. Harvesting such low aboveground biomass is unprofitable. Thin, delicate stems, after being crushed and left on the plantation, will be partially macerated in winter and eventually be introduced into the soil with spring cultivation. Our results are partially consistent with the results obtained using other energy crops by research teams in Germany [30,31].

In 2019, the amount of absorbed $CO_2$ by the coniferous forest in Hyytiälä, Finland was measured by the Institute for Atmospheric and Earth system Research (INAR) at the Faculty of Science of the University of Helsinki (by Kukka-Maaria Kohonen). The result is 4.4 tons of $CO_2$ per hectare per annum (forest over 30 years old)—http://www.hiilipuu.fi/ (accessed on 15 May 2022).

This is less per hectare, per annum, than a one-year sequestration by Sida (five tons C in sandy soil and 15 tons in sandy-loam).

Sida is a well-known energy species with a beneficial effect on the environment [32–34].

In the era of combating excessive carbon dioxide emissions. significant carbon sequestration in the roots and stems of Sida allows the species to be classified as a crop, which is particularly useful in environmental protection. and it can be considered as the future species of Carbon Farming.

### 5. Conclusions

1. Significant carbon sequestration in the root system and above ground biomass allows Sida to be classified as one of the species limiting the concentration of carbon dioxide in the atmosphere.
2. Sida is particularly suitable for cultivation in sandy soils due to the improvement of their mechanical and biological properties.
3. On sandy loam soil, the yield of Sida biomass was several times higher than on sandy soil.
4. Virginia fanpetals (Sida), due its strong Carbon sequestration and environment benefits can be also called The Species for a Changing World.

**Author Contributions:** Conceptualization, R.M. and H.B.; methodology, H.B.; software, E.S.; validation, B.S., D.S. and E.S.; formal analysis, H.B.; investigation, B.S.; resources, E.S.; data curation, B.S.; writing—original draft preparation, H.B.; writing—review and editing, R.M.; visualization, E.S.; supervision, B.S. and S.D; project administration, D.S. All authors have read and agreed to the published version of the manuscript.

**Funding:** This research received no external funding.

**Institutional Review Board Statement:** Not applicable.

**Informed Consent Statement:** Not applicable.

**Data Availability Statement:** Not applicable.

**Conflicts of Interest:** The authors declare no conflict of interest.

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
