# Peer review of "The Sustainable Future of Carbon Farming with Virginia Fanpetals, a Carbon-Negative Species"

_2673-7655, doi:10.3390/crops2030019_

Round 1
Reviewer 1 Report
My comments are as under:
- The abstract must include data regarding the critical finds by the authors in terms of data of important findings.
- The introduction must have a clear hypothesis and significantly develop the second paragraph of this manuscript.
- Overall there is the repetition of the information which could be avoided.
- Check figure ligands; they are carelessly written.
- Discussion should include more information and references related to the relevant and related works.
- Restructure and carefully edit the conclusion section.
Author Response
Thank you.

Reviewer 2 Report
Dear Authors,
I would like to thank you for all your hard and sincere efforts in bringing out this manuscript. I have gone through it and have some queries.
1. What do you mean by carbon sequestration? Will it be proper to say carbon storage? Plant biomass would continuously be mineralized by the microbes and CO2 will be released into the atmosphere, hence in my view the word "sequestration" doesn't fit in this situation.
2. You haven't described the method by which you have determined the earthy fractions in soil? Please include this in your manuscript.
3. In table no. 7 at page no. 5 and line no. 170 you have given the values of fine sand and silt, but it seems that the values are interchanged. You may check it and just change the heading Fine sand (0.02-0.05) with Silt (0.002-0.05) and vice-versa.
4. In the same line (170) as per the data the soil textural class as per USDA system of soil classification, before cultivation is loamy sand but you have mentioned it as sandy. Please check it.
5. In table no. 7 the sand per cent decreased from 76.55 to 62.33 in the first case and in the second case the sand per cent increased from 53.31 to 66.73 do you have any proper explanation for this contrasting effect? Moreover, within such a short time span of three years, such large changes in earthy fractions seem impossible. Please revisit your data, method of analysis and calculation.
6. At line no. 195 you have used the term soil quality. But this term doesn't seem to fit here as soil quality in the summation of all the physical, chemical and biological properties of soil and not just the earthy fractions (Soil Texture).

Author Response
Thank you.

Reviewer 3 Report
The presentation of the article is good and falls in the scope of journal.
However, article need some improvement.
General comments
Line 17. Please change percentage content into percent content
Your analysis showed (Tables) that years (Factor a), and type of soil (Factor b). The data presented in tables showed that the mean of years (Y) is significant and also the mean of soil type (ST) is also significant. However, i not see any report of significant or not significant for the interaction of year x soil type. please add an extra row at the end of all tables, and show that weather Y x Soil types was significant (* , ** , ***) or not significant (ns ).
I suggest, please check your analysis, if the interaction is significant, please make figures for the interaction, and discuss
Author Response
Line 17 and tables are done. Thank you.

Round 2
Reviewer 2 Report
Dear Authors,
I do agree that you have got your samples analysed from a certified agricultural laboratory. But it is your manuscript and it is your responsibility to check the data and results. You should clearly mention the methodology that you have adopted. A considerable portion of your result is based on changes in the composition of earthy fractions (texture) and you haven't described the methodology. It doesn't seem fine. You can check the textural class based on your reported data at the following link.
https://www.nrcs.usda.gov/wps/portal/nrcs/detail/soils/survey/?cid=nrcs142p2_054167
Normally we follow this USDA system for classifying soil texture. If you have adopted some other system you must clearly mention it and describe it.
You say "Sida's root system secretes enzymes and crumbles lumps of loam soil. There is nothing to crush in the sandy soil, and the humus level is constantly increasing (thanks to falling leaves), so sand sticks to humus. In sandy loam, the roots crush the lumps of soil in which the sand is stuck together, hence its greater content. Fine. But when we determine earthy fractions (Texture) we first remove the organic matter by treating the soil samples with hydrogen peroxide.
Ref.: Partide size Analysis by G. W. GlElE Battelle and J. W. BAUllJJER in METHODS· OF SOIL ANALYSIS Part 1 Physical and Mineralogical Methods Second Edition Arnold Klute, Editor (1986) from page no. 383.
So, dear authors, I do feel that a relook at your data and methodology for estimating texture is necessary.
Author Response
Dear Reviewer,
Thank you for your effort to review our work. We really appreciate it. We also appreciate your advice and guidance, which has helped us improve the quality of our work. Here are the detailed references to the notes:
- We agree with the Reviewer's first comment on the methodology. Currently, the research methodology refers to the methods used at the Chemical and Agricultural Station for soil testing, and the relevant literature items are included in the references. We apologize for the lack of a detailed description in the previous version of the work. This part of the work has now been completed. At the Chemical and Agricultural Station, the methodology applicable in certified laboratories in force in Poland, the European Union; and particle fractions (US NCSS, 2014) was applied.
- We are pleased with the understanding of the sand clumping and release process during Sida cultivation, by two different type of soil; that the root system of the Sida in sandy soil has nothing to crush, while the level of humus increases due to the mass of falling leaves, so the sand sticks together with the humus; in sandy loam, in a different manner- the roots crush the soil lumps in which the sand is stuck, hence the change in its content.
- Table 7 is corrected to include the division of soil fractions into three categories (100%). The first column contains a breakdown into 'Soil agronomic categories'. Considering the share of fine particles (<0.02 mm), the tested soils were classified into sandy and sandy loam classes. The soil characteristics in the text were also improved.
- The previous version of Table 7 was presented correctly, except that, that we inadvertently did not indicate that the fourth column contained the designation of fine particles (<0.02 mm), from its whole entirety (column 1+2+3 in fractions = 100%). Now table 7 contains all the information.
- Other comments from the Reviewer and the Editor were also considered.
